# Severity, mortality, long COVID-19, and quality of life: Insights from a cohort study of hospitalized COVID-19 patients during the delta variant predominance period in Thailand

**Pimpinan Khammawan**[1,2], **Aksara Thongprachum**[1], **Kannikar Intawong**[1], **Suwat Chariyalertsak**[1]*

**1** Faculty of Public Health, Chiang Mai University, Chiang Mai, Thailand, **2** Nakornping Hospital, Ministry of Public Health, Chiang Mai, Thailand

* suwat.c@cmu.ac.th

## Abstract

### Background

The COVID-19 pandemic, declared in March 2020, has had significant global impacts, with Thailand reporting over 4.6 million cases and 32,000 fatalities by September 2022. Long COVID, or Post-COVID Conditions (PCC), affects 10–30% of COVID-19 patients globally, with symptoms lasting beyond three months. Common issues include fatigue, brain fog, respiratory problems, and psychological effects such as anxiety and depression. Symptoms can persist regardless of the initial infection severity, and ongoing research continues to refine understanding and management strategies.

To address residual symptoms of COVID-19 during the Delta variant predominance period, a study was conducted from July to December 2021 at a tertiary care hospital in Chiang Mai, Thailand. The study aimed to describe the characteristics of COVID-19 patients, explore the Long COVID symptoms experienced by patients after discharge, and assess their quality of life.

### Methods

The study characterized 604 are moderate to severe COVID-19 patients at Tertiary Care Hospital during the Delta wave in Thailand (July-December 2021), using secondary data from medical records. Confirmed cases were cohort monitored using a Long COVID questionnaire for symptoms, chronic conditions, and social impact a year after discharge. Quality of life was evaluated using the SF-12 questionnaire (SF-12: 12-Item Short Form Survey).

Long COVID, in this study, is defined as the persistence or emergence of one or more physical, psychological, or cognitive symptoms that last for more than 12 weeks

**Data availability statement:** All relevant data are within the paper and its Supporting Information files. Additional data can be made available upon reasonable request.

**Funding:** The author(s) received no specific funding for this work.

**Competing interests:** The authors have declared that no competing interests exist.

after the initial onset of COVID-19 and cannot be explained by alternative diagnoses. This includes, but is not limited to, symptoms such as fatigue, dyspnea, chest pain, cough, cognitive dysfunction ("brain fog"), insomnia, anxiety, or depression.

### Findings

Most patients were Thai (85.9%) and female (57.3%), with obesity common among those aged 18–60 (48.3%). Severe cases and mortality were higher in patients over 60 (30.2%) and unvaccinated patients (60.4%). Severity was related with male gender, older age, lack of antiviral use, and being unvaccinated; overweight status, comorbidities, and abnormal chest x-rays were not significant. Deaths were influenced by gender, age, and antiviral use, but not hospital stay duration, overweight status, comorbidities, or vaccination status. At one-year follow-up, Long COVID symptoms were reported in a small proportion of patients (4.2% shortness of breath, 1.5% chronic cough), mostly in adults and older adults. Other symptoms were rare (<1%) and limited to the 18–60 age group. No severe neurological or systemic symptoms were reported. One-year post-hospitalization, 79.15% had no Long COVID symptoms. Quality of life scores were high (Physical Component Summary: PCS = 48.62, Mental Component Summary: MCS = 50.65).

### Interpretation

This study found a very low prevalence of Long COVID symptoms, which may be due to the severity of the Delta variant leading to higher mortality among patients with severe illness. Those who survived and recovered mostly had moderate symptoms and were predominantly under 60 years of age, which may explain the lower occurrence of Long COVID in this group.

The majority of COVID-19 patients in Chiang Mai experienced moderate symptoms and had a high survival rate. Despite varied long COVID symptoms, most reported good physical and mental health one year after recovery. These findings highlight the resilience of patients and the importance of monitoring long-term health outcomes.

### Introduction

The COVID-19 pandemic, caused by SARS-CoV-2, began in Wuhan, China, in December 2019 and quickly spread globally [1]. The World Health Organization (WHO) declared it a global health emergency on January 30, 2020, and a pandemic on March 11, 2020 [2]. By September 20, 2022, there were 617,573,821 confirmed cases and 6,531,909 deaths worldwide [3]. The virus's rapid spread was facilitated by travel and asymptomatic transmission.

Thailand reported its first COVID-19 case on January 22, 2020, with cases increasing until mid-March 2020. By September 20, 2022, Thailand had 4,633,559 confirmed cases and 32,655 deaths [4,5]. Government policies and public health

measures significantly influenced case trajectories. In Chiang Mai, a key tourist hub, proactive surveillance and public health measures were established early on. As of September 2022, Chiang Mai had reported 36,415 confirmed cases and 380 deaths. COVID-19 patients in Chiang Mai were categorized by severity (Green, Yellow, Red) based on WHO guidelines [6]. COVID-19 patients in Chiang Mai province were placed into three different levels of treatment; green, yellow, and red, according to WHO guidelines. Those patients with mild symptoms were placed in the green category, while patients with moderate symptoms were put into the yellow category, and those with severe symptoms and who needed a ventilator were grouped into the red category [7].

The third wave of COVID-19 led to improved patient management in Thailand, including district and sub-district field hospitals and home care initiatives (Home Isolation and Community Isolation). These measures aimed to provide personalized care and reduce hospital strain. However, many survivors experience Long COVID, with symptoms persisting up to a year post-infection [8–10].

Long COVID refers to persistent effects of COVID-19 lasting weeks or months beyond the initial illness, affecting various organ systems and mental health [11]. Symptoms, appearing 4–12 weeks post-infection, include fatigue, breathing difficulties, cough, insomnia, headache, hair loss, dizziness, anxiety, stress, and chest pain. These can improve, worsen, or recur over time. Also known as post-COVID-19 conditions, post-COVID syndrome, or Long Hauler COVID, affects 10–30% of COVID-19 patients globally [4].

Many COVID-19 patients require close monitoring, there are several gaps, such as limited understanding of disease mechanisms, variability in symptoms, and insufficient population-specific data. Long-term impacts on quality of life and the role of vaccines remain underexplored. Challenges to healthcare systems and factors influencing resilience and recovery also require further research to improve prevention, diagnosis, and disease management.

This study aimed to describe the characteristics of hospitalized COVID-19 patients during the Delta variant predominance and to assess the Long COVID and quality of life of patients after discharge from a tertiary care hospital in Chiang Mai Province.

## Methodology

### Study design

The prospective cohort study aims to explore the clinical characteristics of hospitalized patients infected with COVID-19 during the Delta variant outbreak and examine the factors contributing to Long COVID. It focuses on the prevalence and persistence of symptoms from July to December 2021 after one-year discharge. The study will track both the physical and psychological symptoms from the onset of infection, throughout the disease's course, and up to one year after hospital discharge.

### Study population

The study population consists of COVID-19 patients during the Delta Variant Predominance period, discharged from Tertiary Care Hospital between July and December 2021. All cases were purposively selected, inclusion criteria are a doctor-diagnosed moderate to severe COVID-19, confirmed by a positive RT-PCR result for SAR-CoV-2 infection. Participants must have been hospitalized and discharged for at least one-year, be contactable by phone, able to communicate in Thai, and willing to participate voluntarily.

### Sample size

The selection of cases for this study was purposive, selected all cases concentrating particularly on patients admitted to Nakornping Hospital during the period from July to December 2021. This sample size was 604 cases, study flow of hospitalized COVID-19 patients during the Delta variant predominance period (Fig 1).

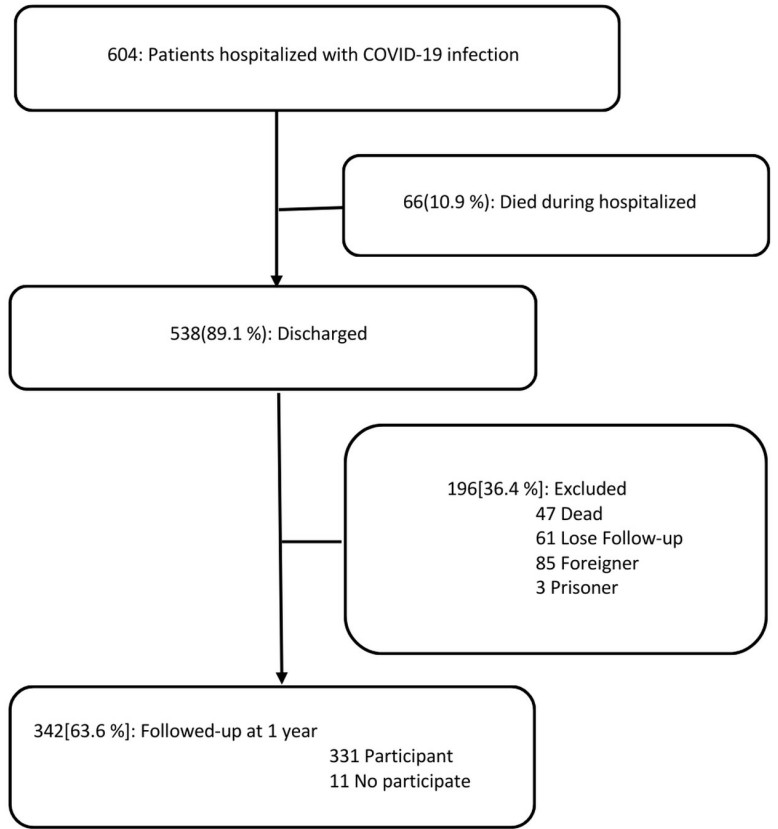

**Fig 1. Flow diagram showing the selection of hospitalized COVID-19 patients during the Delta variant predominance period in Thailand.**

A total of 604 patients were hospitalized between July and December 2021. Among them, 66 (10.9%) died during hospitalization, while 538 (89.1%) were discharged. Of those discharged, 196 (36.4%) were excluded from follow-up due to death (n = 47), loss to follow-up (n = 61), foreign nationality (n = 85), or incarceration (n = 3). The remaining 342 patients (63.6%) were eligible for one-year follow-up, of whom 331 participated and 11 declined as shown in figure 1.The inclusion criteria for this study are as follows: 1. Confirmed positive for SAR-CoV-2 infection through RT-PCR testing, 2. Participants who were willing to participate, 3. Participants reachable by telephone for post-discharge follow-up evaluations and 4. Participants who were able to communication in the Thai language.

## Data sources

This study is divided into two parts. The first part involves collecting secondary data from medical records to describe the characteristics of individuals hospitalized with COVID-19 at Tertiary Care Hospital. This data will include personal information, illness details, treatment specifics, and outcomes, enabling analysis of patient demographics, disease severity, and treatment impact, medical data were accessed for research purposes, 1 December 2022 to 31 January 2023.

The second part involves follow-up evaluations at least one-year after-discharge through 10–15-minute telephone interviews using self-assessment tools. These tools, designed to assess residual symptoms, Long COVID questionnaire will be developed based on a literature review and Quality of Life (QoL), have been carefully reviewed for effectiveness in patient interviews with the SF-12(The SF-12 Short Form: 12-item Health Survey is widely used in global health research) [12–14], the interviewing start at16 February 2023 then finish 10 November 2023.

## Ethical considerations

The study was conducted in Nakornping hospital Chiang Mai, Thailand, and approved by the Ethics Committee of Nakornping hospital (The protocol code is 085/65, and it was approved on November 28, 2022). Informed Consent Statement: Informed consent was obtained from all subjects involved in the study.

Consent to Participate in Research: Participants may consent to join the research verbally by responding or agreeing to the researcher. If they choose to decline participation, they can refuse to provide information through the interview immediately, without any coercion or pressure due to the authority of the medical personnel.

## Data collection

1. A structured questionnaire included closed-ended questions on demographics, socio-economic status, health conditions, behaviors, COVID-19 symptoms, vaccination history, and quality of life. Ethical approval was obtained (NKP No. 085/65).

2. Data collection at Nakornping Hospital comprised two parts: (1) accessing electronic medical records for 604 patients and (2) using an Ethics-approved questionnaire reviewed by experts.

3. The researcher trained four assistants in data collection, interviews, ethical practices, and simulations to ensure understanding.

4. Research assistants conducted interviews, obtaining participant consent for telephone or face-to-face interactions. Parental consent was required for children.

5. Assistants explained the study, assured confidentiality, and emphasized voluntary participation, allowing withdrawal at any time.

6. Interviews were conducted privately via phone or in designated locations, using smartphones/tablets for data recording.

7. Assistants completed the questionnaire and requested permission for follow-ups if needed.

8. Data collection spanned one month for Part 1 and six months for Part 2. Some participants were lost to follow-up due to death post-discharge.

9. Interviews were scheduled at convenient times if participants were unavailable during the day.

10. Data was stored securely on a password-protected Google Drive, accessible only by the researcher.

11. Each participant received a unique code. Data was cleaned in Excel and analyzed in SPSS.

## Statistical analysis

Data were analyzed using SPSS version 26.0 (SPSS Inc., Chicago, IL, USA). Descriptive statistics summarized general characteristics using sums, percentages, means, medians (IQR), and Analytical statistics use Chi-square test ($\chi^2$ test) or Fisher's Exact test is used to compare the proportions of categorical variables, such as gender, age, vaccination status, occupation, ventilator use, and length of hospital stay, with survival status. This test determines whether these variables are associated with mortality rates and survival rates. Long-term physical and psychological symptoms and Quality of Life (QoL) were monitored for one- year post-hospitalization. Multivariate Logistic Regression use to analyzes multiple variables simultaneously while adjusting for confounding factors. Reports Adjusted Odds Ratio (AOR), 95% Confidence Interval (CI), and p-value and analyzed factors associated with Long COVID, including

chronic conditions, with quantitative and qualitative variables examined through means, frequencies, percentages, and hypothesis testing and used to analyze the relationship between independent variables (Gender, Age, BMI, Severity Level, Comorbidity, Occupation) and the outcomes PCS (Physical Component Summary) and MCS (Mental Component Summary). Independent t-test used to compare the mean values of PCS and MCS between two groups of variables, such as: Gender (Male and Female), Disease severity (Moderate and Severe), Comorbidity (Yes and No), Length of hospital stay (≤ 2 weeks and > 2 weeks) [10,15,16].

## Results

### Socio-demographic characteristic of COVID-19 patients

The socio-demographic characteristics of 604 COVID-19 patients from July to December 2021 revealed significant findings. Male patients (258) had a higher mortality rate during admission (27.1%) compared to females (19.9%). Older patients, particularly those over 60, experienced the highest mortality both during admission (34.9%) and post-discharge (22.8%), while no post-discharge deaths occurred in younger patients under 18. Thai patients had a higher mortality rate (26.8%) than non-Thai patients, who reported no deaths. Vaccination status was a critical factor, with unvaccinated individuals showing the highest mortality rates during admission (38.1%) and post-discharge (13.4%), whereas vaccinated patients, even those with a single dose, had no reported deaths. Patients with severe conditions had a 44.8% mortality rate during admission, and those requiring invasive ventilation faced similarly high rates (44.8%). Antiviral treatment was associated with a higher mortality rate (25.2%), and patients with hospital stays longer than two weeks had the highest post-discharge mortality (66.7%). These findings underscore the importance of vaccination, early intervention, and managing illness severity to improve patient outcomes during the pandemic, as shown in Table 1.

### Factors associated with severity of illness in hospitalized COVID-19 patients

The result of study highlights factors associated with illness severity in hospitalized COVID-19 patients. Male patients had higher odds of severe illness (unadjusted OR: 1.614, 95% CI: 1.023–2.547, p=0.040), but this association was not significant after adjustment (adjusted OR: 1.583, p=0.062). Age was a significant factor, with patients over 60 showing much higher odds of severe illness (adjusted OR: 3.569, 95% CI: 2.181–5.839, p<0.001). Overweight patients had slightly increased odds (unadjusted OR: 1.281, p=0.291), but this was not statistically significant. Patients without antiviral treatment had significantly higher odds of severe illness (adjusted OR: 2.547, 95% CI: 1.269–5.115, p=0.009). Longer hospital stays (over two weeks) were associated with increased odds in the unadjusted model (OR: 12.410, p=0.004), but this was not significant after adjustment (adjusted OR: 5.575, p=0.064). Comorbidities were linked to lower odds of severe illness (unadjusted OR: 0.588, p=0.066), and abnormal chest X-rays showed slightly higher odds, though neither finding was statistically significant. Vaccination status was strongly associated with severity, with unvaccinated patients having significantly higher odds of severe illness (adjusted OR: 3.193, 95% CI: 1.795–5.678, p<0.001). These findings emphasize age, vaccination, and antiviral treatment as key factors influencing illness severity, as shown in Table 2.

### Factors associated with survival rates in hospitalized COVID-19 patients

The analysis of factors influencing recovery among hospitalized COVID-19 patients identified several key associations, data from deceased during admission, after discharged and surviving individuals alongside the following factors, Male patients had significantly lower odds of survival compared to females (adjusted OR: 0.381, 95% CI: 0.236–0.614, p<0.001). Age also played a critical role, with patients over 60 showing much lower survival odds than those under 60 (adjusted OR: 0.131, 95% CI: 0.081–0.214, p<0.001). Overweight patients had slightly increased survival odds, but this was not statistically significant. Lack of antiviral treatment was strongly associated with lower survival odds (adjusted OR: 0.332, 95% CI: 0.182–0.606, p<0.001). Longer hospital stays initially appeared to improve survival (unadjusted OR:

**Table1. Socio-demographic characteristic of COVID-19 patients classifies by status on July 2021 to December 2021(n = 604).**

| Variable | Category | Number (%) | | | |
|---|---|---|---|---|---|
| | | Total (n = 604) | Dead during admit (n = 139) | Dead after discharge (n = 49) | Alive (n = 416) |
| Gender | Male | 258 (100.0) | 70 (27.1) | 30 (11.6) | 158 (61.2) |
| | Female | 346 (100.0) | 69 (19.9) | 19 (5.5) | 258 (74.6) |
| Age | <18 year | 122 (100.0) | 15 (12.3) | 0 (0.0) | 107 (87.7) |
| | 18-60 year | 333 (100.0) | 72 (21.6) | 15 (4.5) | 246 (73.9) |
| | > 60 years | 149 (100.0) | 52 (34.9) | 34 (22.8) | 63 (42.3) |
| Nationality | Thai | 519 (100.0) | 139 (26.8) | 47 (9.1) | 333 (64.2) |
| | Not Thai | 85 (100.0) | 0 (0.0) | 2 (2.4) | 83 (97.6) |
| BMI | Underweight (<18.5) | 128 (100.0) | 23 (18.0) | 11 (8.6) | 94 (73.4) |
| | Normal (18.5–24.9) | 243 (100.0) | 57 (23.5) | 22 (9.1) | 164 (67.5) |
| | Obesity (>25) | 233 (100.0) | 59 (25.3) | 16 (6.9) | 158 (67.8) |
| Vaccination | 0 dose | 365 (100.0) | 139 (38.1) | 49 (13.4) | 177 (48.5) |
| | 1 dose | 43 (100.0) | 0 (0.0) | 0 (0.0) | 43 (100.0) |
| | 2 doses | 134 (100.0) | 0 (0.0) | 0 (0.0) | 134 (100.0) |
| | 3 doses | 50 (100.0) | 0 (0.0) | 0 (0.0) | 50 (100.0) |
| | 4 doses | 12 (100.0) | 0 (0.0) | 0 (0.0) | 12 (100.0) |
| Occupation | Government & Health Care Worker | 9 (100.0) | 1 (11.1) | 0 (0.0) | 8 (88.9) |
| | Private | 250 (100.0) | 53 (21.2) | 12 (4.8) | 185 (74.0) |
| | Agriculture | 17 (100.0) | 5 (29.4) | 1 (5.9) | 11 (64.7) |
| | Unemployed | 328 (100.0) | 80 (24.4) | 36 (11.0) | 212 (64.6) |
| Risk group | Cluster | 3 (100.0) | 0 (0.0) | 0 (0.0) | 3 (100.0) |
| | Contact confirm case | 337 (100.0) | 69 (20.5) | 18 (5.3) | 250 (74.2) |
| | House hold contact | 50 (100.0) | 12 (24.0) | 8 (16.0) | 30 (60.0) |
| | Outbreak area, Risk area | 120 (100.0) | 29 (24.2) | 11 (9.2) | 80 (66.7) |
| | Unknown | 91 (100.0) | 28 (30.8) | 12 (13.2) | 51 (56.0) |
| | Re-infection | 3 (100.0) | 1 (33.3) | 0 (0.0) | 2 (66.7) |
| Severity | Moderate | 517 (100.0) | 100 (19.3) | 29 (5.6) | 388 (75.0) |
| | Severe | 87 (100.0) | 39 (44.8) | 20 (23.0) | 28 (32.2) |
| Chest radiograph | Abnormal | 333 (100.0) | 77 (23.1) | 27 (8.1) | 229 (68.8) |
| | Normal | 271 (100.0) | 62 (22.9) | 22 (8.1) | 187 (69.0) |
| Pneumonia | No | 105 (100.0) | 32 (30.5) | 9 (8.6) | 64 (61.0) |
| | Yes | 499 (100.0) | 107 (21.4) | 40 (8.0) | 352 (70.5) |
| Antivirus | Use antivirus | 408 (100.0) | 103 (25.2) | 43 (10.5) | 262 (64.2) |
| | No use antivirus | 196 (100.0) | 36 (18.4) | 6 (3.1) | 154 (78.6) |
| Intubation | Non-Invasive | 517 (100.0) | 100 (19.3) | 29 (5.6) | 388 (75.0) |
| | Invasive ventilator | 87 (100.0) | 39 (44.8) | 20 (23.0) | 28 (32.2) |
| Length of stay | ≤ 2week | 598 (100.0) | 138 (23.1) | 45 (7.5) | 415 (69.4) |
| | > 2weeks | 6 (100.0) | 1 (16.7) | 4 (66.7) | 1 (16.7) |

22.685, p = 0.005), but the effect was not significant after adjustment. Comorbidities were linked to higher survival odds in the unadjusted model but lost significance after adjustment. Abnormal chest X-rays showed no significant association with survival, and vaccination status indicated no measurable effect. These findings highlight the importance of gender, age, and antiviral treatment in influencing recovery outcomes, as shown in Table 3.

**Table 2. Factors associated with severity of illness in hospitalized COVID-19 patients.**

| Variable | Category | Unadjusted Odd Ratio (95% CI) | p-value | Adjusted Odd Ratio (95% CI) | p-value |
|---|---|---|---|---|---|
| Gender | Female | 1 | | 1 | |
| | Male | 1.614(1.023-2.547) | 0.040 | 1.583(0.977-2.565) | 0.062 |
| Age | <18–60 years | 1 | | 1 | |
| | >60 years | 4.255(2.653-6.823) | <0.001 | 3.569(2.181-5.839) | <0.001 |
| BMI | Normal | 1 | | | |
| | Abnormal (Overweight) | 1.281(0.809-2.027) | 0.291 | | |
| Antivirus use | Yes | 1 | | 1 | |
| | No | 3.850(1.995-7.429) | <0.001 | 2.547(1.269-5.115) | 0.009 |
| Length of stay | Length of stay ≤ 2 week | 1 | | 1 | |
| | Length of stay > 2 week | 12.410(2.237-68.828) | 0.004 | 5.575(0.906-34.287) | 0.064 |
| Comorbidity | No | 1 | | | |
| | Yes | 0.588(0.334-1.036) | 0.066 | | |
| Chest radiograph | Normal | 1 | | | |
| | Abnormal | 1.319(0.830-2.099) | 0.242 | | |
| Vaccinated | Yes | 1 | | 1 | |
| | No | 2.862(1.656-4.947) | <0.001 | 3.193(1.795-5.678) | <0.001 |

**Table 3. Factors associated with survival rates in hospitalized COVID-19 patients.**

| Variable | Category | Unadjusted Odd Ratio (95% CI) | p-value | Adjusted Odd Ratio (95% CI) | p-value |
|---|---|---|---|---|---|
| Gender | Female | 1 | | 1 | |
| | Male | 0.418(0.275-0.635) | <0.001 | 0.381(0.236-0.614) | <0.001 |
| Age | <18–60 years | 1 | | 1 | |
| | >60 years | 0.132(0.081-0.215) | <0.001 | 0.131(0.081-0.214) | <0.001 |
| BMI | Normal | 1 | | | |
| | Abnormal (Overweight) | 1.076(0.705-1.642) | 0.733 | | |
| Antivirus use | Yes | 1 | | 1 | |
| | No | 0.309(0.179-0.534) | <0.001 | 0.332(0.182-0.606) | <0.001 |
| Length of stay | Length of stay ≤ 2 week | 1 | | 1 | |
| | Length of stay > 2 week | 22.685(2.624-196.140) | 0.005 | 0.143(0.15-1.357) | 0.090 |
| Comorbidity | No | 1 | | 1 | |
| | Yes | 2.665(1.628-4.361) | <0.001 | 0.700(0.394-1.242) | 0.223 |
| Chest radiograph | Normal | 1 | | | |
| | Abnormal | 0.928(0.614-1.401) | 0.721 | | |
| Vaccinated | No | 1 | | | |
| | YES | 0.000(0.000) | 0.994 | | |

## Long COVID after discharge at one-year follow up

The Table 4 presents symptoms experienced by COVID-19 patients in different age groups at one-year follow-up post-discharge among 331 patients. Shortness of breath is the most commonly reported symptom, affecting 14 patients (4.2%), with 1.1% in the < 18 age group, 5.0% in the 18–60 age group, and 7.0% in the > 60 age group. Chronic cough was reported by 5 patients (1.5%) overall, including 1.1% in the < 18 group, 1.7% in the 18–60 group, and 1.8% in the > 60

**Table 4. Long COVID classify by age group after discharge at one-year follow up (n = 331).**

| Variable | Number (%) | | | |
|---|---|---|---|---|
| | Total | <18 year | 18-60 year | >60 year |
| | (n = 331) | (n = 94) | (n = 180) | (n = 57) |
| Palpitation | 1 (0.3) | 0 (0.0) | 1 (0.6) | 0 (0.0) |
| Seizures | 0 (0.0) | 0 (0.0) | 0 (0.0) | 0 (0.0) |
| Poor sleep | 2 (0.6) | 0 (0.0) | 2 (1.1) | 0 (0.0) |
| Brain fog | 2 (0.6) | 0 (0.0) | 2 (1.1) | 0 (0.0) |
| Paraplegia | 0 (0.0) | 0 (0.0) | 0 (0.0) | 0 (0.0) |
| Severe Headache | 0 (0.0) | 0 (0.0) | 0 (0.0) | 0 (0.0) |
| Vertigo | 1 (0.3) | 0 (0.0) | 1 (0.6) | 0 (0.0) |
| Loss of taste and smell | 1 (0.3) | 0 (0.0) | 1 (0.6) | 0 (0.0) |
| Shortness of breath | 14 (4.2) | 1 (1.1) | 9 (5.0) | 4 (7.0) |
| Chronic cough | 5 (1.5) | 1 (1.1) | 3 (1.7) | 1 (1.8) |
| Change in voice | 0 (0.0) | 0 (0.0) | 0 (0.0) | 0 (0.0) |
| Muscle or joint aches | 0 (0.0) | 0 (0.0) | 0 (0.0) | 0 (0.0) |
| Rash | 0 (0.0) | 0 (0.0) | 0 (0.0) | 0 (0.0) |
| Hair Loss | 1 (0.3) | 0 (0.0) | 1 (0.0) | 0 (0.0) |
| Change in menstrual cycle | 0 (0.0) | 0 (0.0) | 0 (0.0) | 0 (0.0) |
| Sore throat | 0 (0.0) | 0 (0.0) | 0 (0.0) | 0 (0.0) |
| Chest congestion | 0 (0.0) | 0 (0.0) | 0 (0.0) | 0 (0.0) |
| Chest pain | 0 (0.0) | 0 (0.0) | 0 (0.0) | 0 (0.0) |
| Pain in abdomen | 1 (0.3) | 0 (0.0) | 1 (0.6) | 0 (0.0) |
| Constipation | 2 (0.6) | 0 (0.0) | 2 (1.1) | 0 (0.0) |
| Nausea/vomiting | 0 (0.0) | 0 (0.0) | 0 (0.0) | 0 (0.0) |
| Diarrhea | 1 (0.3) | 0 (0.0) | 1 (0.6) | 0 (0.0) |

group. Other symptoms, such as palpitations, vertigo, loss of taste and smell, and hair loss, were each reported by only 1 patient (0.3%), all in the 18–60 age group. Additional symptoms, including poor sleep, brain fog, and constipation, were reported by 2 patients each (0.6%) in the 18–60 group. No cases were recorded for seizures, paraplegia, severe headache, change in voice, muscle or joint aches, rash, menstrual cycle changes, sore throat, chest congestion, chest pain, nausea/vomiting, or changes in the menstrual cycle across all age groups. Overall, the occurrence of Long COVID symptoms is relatively low, with the majority of symptoms affecting a small proportion of patients, primarily in the 18–60 and >60 age groups.

## Distribution of long COVID at one-year follow-up

The distribution of long COVID symptoms at the one-year mark shows that out of 331 patients, 262 (79.15%) did not experience any long COVID symptoms. A total of 52 patients (15.71%) reported having one symptom, 14 patients (4.23%) experienced two symptoms, and only 3 patients (0.91%) reported having more than two long COVID symptoms. This data indicates that the majority of patients had either no symptoms or only mild, isolated symptoms at the one-year follow-up.

## Comprehensive analysis of physical and mental component summaries (PCS and MCS) Across Demographic and Clinical Subgroups of COVID-19 Patients

A comprehensive analysis of 331 COVID-19 patients revealed significant associations between demographic and clinical factors and Physical (PCS) and Mental (MCS) Component Summary scores. Gender showed no significant

differences in PCS or MCS scores (p=0.639). Age significantly impacted scores, with patients under 18 reporting the highest PCS (51.06±2.54) and MCS (50.94±1.13), and those over 60 having lower PCS (45.25±5.82) but comparable MCS (50.89±3.27) (p<0.001). Severity of illness and comorbidity influenced PCS, with severe cases (49.02±3.68) and patients with comorbidities (49.24±4.09) scoring higher than others (p<0.001). Occupational groups also differed significantly (p=0.024), with higher PCS among government/healthcare workers (49.43±4.43). Length of hospital stay was notably significant, with shorter stays (<2 weeks) associated with better PCS (48.76±4.54) and MCS (50.58±3.06), while longer stays drastically reduced PCS (6.69±5.47) (p<0.001). Overall, age, illness severity, comorbidities, occupation, and hospital stay length significantly affected PCS and MCS scores., as shown in Table 5.

### Factors Associated with Physical and Mental Component Summary (PCS and MCS) Scores in Hospitalized COVID-19 Patients

The Table 6 presents the odds ratios (OR), confidence intervals (CI), and p-values for factors influencing Physical Component Summary (PCS) and Mental Component Summary (MCS) scores in hospitalized COVID-19 patients. PCS (Physical Component Summary): Age (<18–60 years) is significantly associated with higher PCS scores (OR = 5.829, p<0.001), indicating better physical outcomes compared to those >60 years. Normal BMI is linked to better PCS scores (OR = 1.697, p=0.027) compared to overweight individuals. Absence of comorbidities significantly improves PCS scores (OR = 5.692, p<0.001). Employed individuals have lower PCS scores compared to unemployed ones (OR = 0.414, p<0.001). Gender (p=0.925) and severity level (p=0.687) are not significantly associated with PCS. MCS (Mental Component Summary): No significant associations are found between MCS and gender (p=0.767), age (p=0.432), BMI (p=0.287), comorbidity (p=0.228), or employment status (p=0.25). Moderate severity cases show a non-significant trend toward higher MCS scores (OR = 2.220, p=0.099).

**Table 5. Comprehensive analysis of physical and mental component summaries (PCS and MCS) across demographic and clinical subgroups of COVID-19 patients.**

| Category | Subcategory | n | PCS Mean±SD | p-value | MCS Mean±SD | p-value |
|---|---|---|---|---|---|---|
| Gender | Male | 133 | 48.80±4.27 | 0.639 | 50.64±3.16 | 0.639 |
| | Female | 198 | 48.50±4.97 | | 50.66±2.96 | |
| Age | <18 years | 94 | 51.06±2.54 | <0.001 | 50.94±1.13 | <0.001 |
| | 18-60 years | 180 | 48.41±4.46 | | 50.43±3.58 | |
| | >60 years | 57 | 45.25±5.82 | | 50.89±3.27 | |
| Severity level | Moderate | 312 | 48.59±4.75 | <0.001 | 50.72±2.96 | <0.001 |
| | Severe | 19 | 49.02±3.68 | | 49.53±4.00 | |
| Comorbidity | Yes | 282 | 49.24±4.09 | <0.001 | 50.69±2.56 | <0.001 |
| | No | 49 | 45.02±6.15 | | 50.41±4.98 | |
| BMI Category | Underweight (<18.5) | 81 | 50.06±4.14 | 0.891 | 50.90±1.88 | 0.891 |
| | Normal (18.5–24.9) | 122 | 48.34±4.51 | | 50.59±3.42 | |
| | Obesity (>25) | 128 | 47.97±5.02 | | 50.55±3.25 | |
| Occupation | Government & Health Care Worker | 8 | 49.43±4.43 | 0.024 | 50.40±1.74 | 0.024 |
| | Private | 154 | 47.65±4.95 | | 50.72±3.50 | |
| | Agriculture | 11 | 49.05±4.19 | | 50.56±2.27 | |
| | Unemployed | 158 | 49.49±4.32 | | 50.60±2.64 | |
| Length of Stay | ≤ 2 weeks | 316 | 48.76±4.54 | <0.001 | 50.58±3.06 | <0.001 |
| | > 2 weeks | 15 | 6.69±5.47 | | 52.33±1.81 | |
| | | | | | | |

**Table 6. Factors associated with physical and mental component summary (PCS and MCS) scores in hospitalized COVID-19 patients.**

| Variable | Category | PCS | | MCS | |
|---|---|---|---|---|---|
| | | Odd Ratio(95%CI) | P-value | Odd Ratio(95%CI) | P-value |
| Gender | Male | 1.023(0.640-1.635) | 0.925 | 1.079(0.651-1.788) | 0.767 |
| | Female | 1 | | 1 | |
| Age | <18–60 years | 5.829(3.154-10.770) | <0.001 | 1.287(0.686-2.418) | 0.432 |
| | >60 years | 1 | | 1 | |
| BMI | Normal | 1.697(1.063-2.709) | 0.027 | 1.313(0.796-2.167) | 0.287 |
| | Abnormal (Overweight) | 1 | | 1 | |
| Severity Level | Moderate | 1.219(0.466-3.189) | 0.687 | 2.220(0.862-5.719) | 0.099 |
| | Severe | 1 | | 1 | |
| Comorbidity | No | 5.692(2.963-10.936) | <0.001 | 0.668(0.347-1.287) | 0.228 |
| | Yes | 1 | | 1 | |
| Occupation | Employed | 0.414(0.256-0.668) | <0.001 | 0.747(0.454-1.228) | 0.250 |
| | Unemployed | 1 | | 1 | |

## Discussion

This cohort study aligns with and extends findings from recent research on the impact of the Delta variant, long COVID prevalence, and health-related quality of life (HR-QoL) in COVID-19 patients.

### Mortality and risk factors

The findings that males, older adults, and unvaccinated individuals had higher mortality rates match those of Hernández-Aceituno A., et al [17]. Their study confirms that the Delta variant presented higher mortality risks compared to Omicron, particularly in unvaccinated and elderly patients [18]. Similarly, O'Kelly et al [19] emphasized that older adults are at the highest risk for severe disease, underscoring the need for vaccination and targeted care.

These studies reinforce that early intervention, vaccination, and special attention to high-risk groups are essential strategies to reduce mortality rates and improve patient outcomes. This study is supported by another research of Chari-yalertsak S' study, the use of heterologous vaccination schedules during the Delta and Omicron outbreaks. The findings highlight the benefits of receiving four vaccine doses, showing a high likelihood of preventing infection, death, and severe COVID-19 outcomes [20].

### Long COVID and symptom persistence

The low prevalence of severe long COVID in this study resonates with results from Agergaard et al. (2023), who found that most patients infected with earlier variants, including Delta, reported only mild or temporary symptoms 1.5 years post-infection [18]. O'Kelly et al. also found a limited prevalence of debilitating long COVID symptoms at the one-year mark, affirming your observation that many patients recover well over time [19], both studies highlight the importance of follow-up care to monitor symptoms, but also offer a positive outlook that severe long COVID cases are not widespread.

### Impact of vaccination and antiviral use

The findings about the effectiveness of vaccination and early antiviral intervention align with those of Hernández-Aceituno et al., who demonstrated that vaccinated patients experienced significantly fewer severe outcomes compared to unvaccinated individuals across all variants studied [17]. Kim et al. (2023) similarly observed that vaccination and pharmaceutical

interventions reduced the severity of long-term symptoms, particularly in high-risk patients [21]. These findings reinforce the need for continued vaccination efforts, booster programs, and timely use of antivirals to prevent severe illness and improve recovery outcomes.

### Health-related quality of life

The HR-QoL findings from study, particularly that younger patients have better physical health scores, mirror those in Kim et al.'s study. They reported that younger patients tended to recover more quickly and had better quality of life two years after acute COVID-19 infection [21]. O'Kelly et al. also found consistent mental health outcomes across age groups, supporting your conclusion that mental health impacts are more variable in severe cases but stabilize over time [19].

The results underscore the importance of personalized rehabilitation programs, particularly for older adults and those who experienced severe illness, to ensure long-term improvements in both physical and mental health outcomes. Physical rehabilitation should focus on older adults, overweight individuals, and those with comorbidities, while workplace policies should support employed patients during recovery. Further research is needed to explore psychosocial influences on MCS and clarify the impact of hospital length of stay on recovery.

### Strengths and limitations

This study has some limitations. First, the sample size was relatively small, and the cohort was restricted to a specific region, potentially limiting the generalizability of the findings. Second, the study relied on self-reported data for Long COVID symptoms, which may be subject to recall bias. Third, baseline quality of life for patients prior to their COVID-19 diagnosis is unavailable, as it was not assessed during the initial admission or pre-infection period. This limitation restricts the ability to compare pre- and post-infection quality of life or to accurately evaluate the impact of COVID-19 and Long COVID on overall patient well-being. Future studies should consider including baseline assessments to provide a more comprehensive understanding of these effects.

### Conclusions and forward view

This study highlights key factors influencing COVID-19 severity, mortality, and long-term outcomes. Age, gender, and antiviral treatments significantly impacted disease outcomes, emphasizing the need for timely interventions. High quality of life scores one-year post-discharge suggest effective post-discharge care. Comorbidities and initial infection severity play complex roles in recovery, indicating the need for personalized management strategies. These findings underscore the critical role of vaccination, the importance of early and continuous care, and the necessity for ongoing research to understand and address COVID-19's long-term impacts.

### Supporting information

**S1 File. Data. Raw dataset (de-identified) used for analysis.**
(XLSX)

### Acknowledgments

Acknowledgments are extended to the Ethics Committee and the administration of Nakornping Hospital for authorizing the use of data in this study. Gratitude is also expressed to the patients and their families for their cooperation. Additionally, appreciation is conveyed to the advisors from the Faculty of Public Health at Chiang Mai University for their valuable insights and guidance throughout the research endeavor.

## Author contributions

**Conceptualization:** Pimpinan Khammawan, Aksara Thongprachum, Kannikar Intawong, Suwat Chariyalertsak.

**Data curation:** Pimpinan Khammawan.

**Formal analysis:** Pimpinan Khammawan.

**Funding acquisition:** Pimpinan Khammawan.

**Investigation:** Pimpinan Khammawan.

**Methodology:** Pimpinan Khammawan, Kannikar Intawong.

**Project administration:** Pimpinan Khammawan.

**Resources:** Pimpinan Khammawan.

**Software:** Pimpinan Khammawan, Kannikar Intawong.

**Supervision:** Pimpinan Khammawan, Aksara Thongprachum, Kannikar Intawong, Suwat Chariyalertsak.

**Validation:** Pimpinan Khammawan.

**Visualization:** Pimpinan Khammawan.

**Writing – original draft:** Pimpinan Khammawan.

**Writing – review & editing:** Pimpinan Khammawan.

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
