## [Decision Letter · Decision Letter 0]

Dear Dr. khammawan,

Thank you for submitting your manuscript to PLOS ONE. After careful consideration, we feel that it has merit but does not fully meet PLOS ONE’s publication criteria as it currently stands. Therefore, we invite you to submit a revised version of the manuscript that addresses the points raised during the review process.

We look forward to receiving your revised manuscript.

Kind regards,

Oyelola A. Adegboye, PhD

Academic Editor

PLOS ONE

Journal Requirements:

For additional information about PLOS ONE ethical requirements for human subjects research, please refer to http://journals.plos.org/plosone/s/submission-guidelines#loc-human-subjects-research .

3. We note that your Data Availability Statement is currently as follows: “All relevant data are within the manuscript and in Supporting Information files.”

Please confirm at this time whether or not your submission contains all raw data required to replicate the results of your study. Authors must share the “minimal data set” for their submission. PLOS defines the minimal data set to consist of the data required to replicate all study findings reported in the article, as well as related metadata and methods (https://journals.plos.org/plosone/s/data-availability#loc-minimal-data-set-definition). For example, authors should submit the following data: - The values behind the means, standard deviations and other measures reported; - The values used to build graphs; - The points extracted from images for analysis. Authors do not need to submit their entire data set if only a portion of the data was used in the reported study. If your submission does not contain these data, please either upload them as Supporting Information files or deposit them to a stable, public repository and provide us with the relevant URLs, DOIs, or accession numbers. For a list of recommended repositories, please see https://journals.plos.org/plosone/s/recommended-repositories. If there are ethical or legal restrictions on sharing a de-identified data set, please explain them in detail (e.g., data contain potentially sensitive information, data are owned by a third-party organization, etc.) and who has imposed them (e.g., an ethics committee). Please also provide contact information for a data access committee, ethics committee, or other institutional body to which data requests may be sent. If data are owned by a third party, please indicate how others may request data access.

5. Please amend the manuscript submission data (via Edit Submission) to include author Kannikar Intawong.

6. Please ensure that you refer to Figure 1 in your text as, if accepted, production will need this reference to link the reader to the figure.

7. We note you have included a table to which you do not refer in the text of your manuscript. Please ensure that you refer to Table 4 in your text; if accepted, production will need this reference to link the reader to the Table.

Reviewers' comments:

Reviewer's Responses to Questions

**Comments to the Author**

1. Is the manuscript technically sound, and do the data support the conclusions?

Reviewer #1: Yes

Reviewer #2: Partly

Reviewer #3: Yes

2. Has the statistical analysis been performed appropriately and rigorously?

Reviewer #1: Yes

Reviewer #2: No

Reviewer #3: Yes

3. Have the authors made all data underlying the findings in their manuscript fully available?

Reviewer #1: Yes

Reviewer #2: Yes

Reviewer #3: No

4. Is the manuscript presented in an intelligible fashion and written in standard English?

Reviewer #1: No

Reviewer #2: No

Reviewer #3: Yes

Reviewer #1: There are grammatical mistakes in the manuscript. I recommend the authors to undertake a thorough review of the manuscript taking the help of someone fluent in the English language to correct them. This will make the submission more articulate.

Reviewer #2: Abstract: The abstract should be revised to include information on mortality and the clinical characteristics of Long COVID. Additionally, the abbreviations for quality of life (PCS, MCS) should be clarified.

Introduction: The authors should address the research gap and clearly state the objective of the study.

Method: What is the study population? The authors state, “the study population consists of moderate to severe COVID-19 patients discharged from Nakornping Hospital between July and December 2021.” Please ensure the data is consistent with the results, particularly regarding patients who died during admission. All research tools should be described in greater detail, including their interpretation and validation. Who conducted the interviews, and how many interviewers were involved? The statistical analysis section should clearly outline the methods used for comparing QoL scores. The logistic regression exploring associations with Long COVID needs verification, as the authors' description lacks clarity and does not include the results. Please confirm whether multivariate analysis or multivariable analysis was used for this study.

Results: The results section requires significant revision. The authors should clarify the number or sample size used for the analysis. What severity levels are presented: moderate vs. severe? Were patients who died during admission included in the analysis? Which death cases were used in Table 3—those who died during admission or after discharge? The odds ratios (OR) for each factor and the reporting of p-values should be checked for accuracy, particularly for variables such as age and vaccination. Table 4 should also be rechecked, as some variables report p-values inappropriately, and the scales or characteristics of the data that are appropriate for statistical analysis, such as length of stay, should be reconsidered.

Discussion: The discussion cannot provide meaningful comments at this stage because the study results need to be thoroughly checked.

Reviewer #3: Review of the article: Severity, Mortality, Long COVID-19, and Quality of Life: Insights from a Cohort Study of Hospitalized COVID-19 Patients in a Thai Tertiary Care Hospital

Thank you very much for allowing us to review this interesting article. The study was reviewed according to the STROBE guidelines. It is very well written and organized.

It is important to mention that the calculation of sample size should be included in the study, as analytical calculations were performed regarding the factors that impacted severity, mortality, and quality of life. The main objective is not highlighted in the abstract nor in the main document. Additionally, since baseline quality of life was not evaluated, and no scale was applied to assess resilience, it cannot be asserted with certainty that the patients were resilient.

I would like to ask whether the included patients were consecutive admissions of COVID patients, and the indications for invasive mechanical ventilation, as it is striking that the mortality of ventilated patients reported in the study (11%) is considerably lower than what is reported in the global literature, where most series reported more than 50% and even up to 80% in this type of patient, considering that the main indication for mechanical ventilation was ARDS (acute respiratory distress syndrome).

**Do you want your identity to be public for this peer review?** For information about this choice, including consent withdrawal, please see our Privacy Policy

Reviewer #1: **Yes: ** Neeraj Kumar Gupta

Reviewer #2: No

Reviewer #3: **Yes: ** David Rene Rodriguez Lima

---

## [Author Response · Author response to Decision Letter 1]

12 Dec 2024

To the Editor and All Concerned,

I sincerely thank you for the valuable suggestions and have made revisions based on all the recommendations provided, as detailed in the attached document. I look forward to receiving further feedback and am willing to make additional adjustments should there still be any shortcomings in this journal submission.

sincerely

Pimpinan khammawan

---

## [Decision Letter · Decision Letter 1]

Dear Dr. khammawan,

The reviewers have expressed concerns that some of their previous comments were not sufficiently addressed. Please carefully review both the previous and new comments, ensuring that each is thoroughly addressed. In your response, provide specific and detailed explanations to directly address the reviewers' concerns.

We look forward to receiving your revised manuscript.

Kind regards,

Oyelola A. Adegboye, PhD

Academic Editor

PLOS ONE

Reviewers' comments:

Reviewer's Responses to Questions

**Comments to the Author**

Reviewer #1: All comments have been addressed

Reviewer #3: (No Response)

2. Is the manuscript technically sound, and do the data support the conclusions?

Reviewer #1: Yes

Reviewer #3: Yes

3. Has the statistical analysis been performed appropriately and rigorously?

Reviewer #1: Yes

Reviewer #3: No

4. Have the authors made all data underlying the findings in their manuscript fully available?

Reviewer #1: Yes

Reviewer #3: Yes

5. Is the manuscript presented in an intelligible fashion and written in standard English?

Reviewer #1: Yes

Reviewer #3: Yes

Reviewer #1: (No Response)

Reviewer #3: Thank you for allowing me to review this interesting work. I have some considerations:

Abstract

There is a significant grammatical and writing error in the "Methods" section.

The study characterized 604 moderate to severe COVID-19 patients treated at a Tertiary Care Hospital during the Delta wave in Thailand (July–December 2021) by analyzing secondary data from medical records. Confirmed cases were monitored as a cohort using a Long COVID questionnaire to track symptoms, chronic conditions, and assess social impact a year after discharge. Quality of life, including physical and mental health, was evaluated using the SF-12 questionnaire.

The first time an acronym is used, its full meaning must be provided.

Introduction

The introduction must conclude with the clearly stated objective of the research.

Statistical Analysis

Descriptive statistics are well presented, but it is unclear why and in which part of the results the chi-square or Fisher's exact test was used for comparing qualitative variables and among which groups.

Analytical results are presented, not just descriptive ones, but there is no mention of the sample size calculation apart from its inclusion in the selection flowchart.

Results

In Table 4, which shows the analysis of mental and physical components, it seems that a t-test was used to compare the mean score in each category. However, this is not reflected in the statistical analysis section. Additionally, there is no multivariate analysis to determine which independent factors were associated with worse quality-of-life scores. Was this not performed, or was it simply not shown?

**Do you want your identity to be public for this peer review?** For information about this choice, including consent withdrawal, please see our Privacy Policy

Reviewer #1: No

Reviewer #3: **Yes: ** David Rene Rodriguez LiMA

---

## [Author Response · Author response to Decision Letter 2]

26 Feb 2025

Reviewer #3:

Abstract

There is a significant grammatical and writing error in the "Methods" section.

The study characterized 604 moderate to severe COVID-19 patients treated at a Tertiary Care Hospital during the Delta wave in Thailand (July–December 2021) by analyzing secondary data from medical records. Confirmed cases were monitored as a cohort using a Long COVID questionnaire to track symptoms, chronic conditions, and assess social impact a year after discharge. Quality of life, including physical and mental health, was evaluated using the SF-12 questionnaire.

The first time an acronym is used, its full meaning must be provided.

Reviewer #3:

Abstract

(Methods)

The study characterized 604 are moderate to severe COVID-19 patients at Tertiary Care Hospital during the Delta wave in Thailand (July-December 2021), using secondary data from medical records. Confirmed cases were cohort monitored using a Long COVID questionnaire for symptoms, chronic conditions, and social impact a year after discharge. Quality of life was evaluated using the SF-12 questionnaire (SF-12: 12-Item Short Form Survey).

Page 2

Abstract

(Methods)

Introduction

The introduction must conclude with the clearly stated objective of the research.

This study have 3 major objective:

This study aimed to investigate the factors associated with Long COVID among hospitalized patients during the predominance of the Delta variant, to describe the characteristics of hospitalized COVID-19 patients during the Delta variant predominance and to assess the quality of life of patients after discharge from a tertiary care hospital in Chiang Mai Province.

Page 3

Introduction

(paragraph 5th)

Statistical Analysis

Descriptive statistics are well presented, but it is unclear why and in which part of the results the chi-square or Fisher's exact test was used for comparing qualitative variables and among which groups.

Analytical results are presented, not just descriptive ones, but there is no mention of the sample size calculation apart from its inclusion in the selection flowchart.

Sample Size:

The selection of cases for this study was purposive, selected all cases concentrating particularly on patients admitted to Nakornping Hospital during the period from July to December 2021. This sample size was 604 cases, as shown in figure 1.

The inclusion criteria for this study are as follows: 1. Confirmed positive for SAR-CoV-2 infection through RT-PCR testing, 2. Participants who were willing to participate, 3. Participants reachable by telephone for post-discharge follow-up evaluations and 4. Participants who were able to communication in the Thai language.

Page 4

Sample Size

Statistics:

Data were analyzed using SPSS version 26.0 (SPSS Inc., Chicago, IL, USA). Descriptive statistics summarized general characteristics using sums, percentages, means, medians (IQR), and Analytical statistics use Chi-square test (χ² test) or Fisher’s Exact test is used to compare the proportions of categorical variables, such as gender, age, vaccination status, occupation, ventilator use, and length of hospital stay, with survival status. This test determines whether these variables are associated with mortality rates and survival rates. Long-term physical and psychological symptoms and Quality of Life (QoL) were monitored for one- year post-hospitalization. Multivariate Logistic Regression use to analyzes multiple variables simultaneously while adjusting for confounding factors. Reports Adjusted Odds Ratio (AOR), 95% Confidence Interval (CI), and p-value and analyzed factors associated with Long COVID, including chronic conditions, with quantitative and qualitative variables examined through means, frequencies, percentages, and hypothesis testing and used to analyze the relationship between independent variables (Gender, Age, BMI, Severity Level, Comorbidity, Occupation) and the outcomes PCS (Physical Component Summary) and MCS (Mental Component Summary). Independent t-test used to compare the mean values of PCS and MCS between two groups of variables, such as: Gender (Male and Female), Disease severity (Moderate and Severe), Comorbidity (Yes and No), Length of hospital stay (≤ 2 weeks and > 2 weeks).

Page 5

Statistic

Results

In Table 4, which shows the analysis of mental and physical components, it seems that a t-test was used to compare the mean score in each category. However, this is not reflected in the statistical analysis section. Additionally, there is no multivariate analysis to determine which independent factors were associated with worse quality-of-life scores. Was this not performed, or was it simply not shown?

Independent t-test used to compare the mean values of PCS and MCS

multivariate analysis to determine which independent factors were associated with worse quality-of-life scores shown in the Table 5: Factors Associated with Physical and Mental Component Summary (PCS and MCS) Scores in Hospitalized COVID-19 Patients

Page 5

Statistic

Page 7&13

- t-test

- multivariate analysis

---

## [Decision Letter · Decision Letter 2]

Dear Dr. khammawan,

Thank you for submitting your manuscript to PLOS ONE. After careful consideration, we feel that it has merit but does not fully meet PLOS ONE’s publication criteria as it currently stands. Therefore, we invite you to submit a revised version of the manuscript that addresses the points raised during the review process.

We look forward to receiving your revised manuscript.

Kind regards,

Oyelola A. Adegboye, PhD

Academic Editor

PLOS ONE

Reviewers' comments:

Reviewer's Responses to Questions

**Comments to the Author**

Reviewer #1: (No Response)

Reviewer #3: All comments have been addressed

2. Is the manuscript technically sound, and do the data support the conclusions?

Reviewer #1: Yes

Reviewer #3: Partly

3. Has the statistical analysis been performed appropriately and rigorously?

Reviewer #1: Yes

Reviewer #3: Yes

4. Have the authors made all data underlying the findings in their manuscript fully available?

Reviewer #1: Yes

Reviewer #3: Yes

5. Is the manuscript presented in an intelligible fashion and written in standard English?

Reviewer #1: Yes

Reviewer #3: Yes

Reviewer #1: (No Response)

Reviewer #3: Second Review

Abstract

Background(Abstract)

“The data collected from screening forms and patient history files were analyzed to highlight the clinical characteristics of Long COVID and its implications for patient well-being.”

I believe this sentence is unnecessary in the background section of the abstract. The paragraph reads more clearly if it ends with the objectives. Moreover, this part belongs to the methods section.

Findings (Abstract)

In the results reported in the abstract, the description begins with the population, followed by findings on factors associated with severity and mortality. However, the results regarding factors associated with Long COVID — which is the main objective — are not presented. This must be included, as it is the main focus and is not clear when reading the abstract.

This final part of the findings is good and aligns well with the secondary objectives of the study:

“One year post-hospitalization, 79.15% had no Long COVID symptoms. Quality of life scores were high (Physical Component Summary: PCS = 48.62, Mental Component Summary: MCS = 50.65)”

Interpretation (Abstract)

In the interpretation section of the article, there should also be alignment with the main objective, which is to identify factors associated with Long COVID. This should be addressed in the conclusion.

This final part of the interpretation is appropriate and relates to the study's secondary objectives:

“Despite varied long COVID symptoms, most reported good physical and mental health one year after recovery. These findings highlight the resilience of patients and the importance of monitoring long-term health outcomes.”

Introduction

The introduction now more clearly states the objectives; however, it is advisable to present the objectives at the end of the introduction. The last paragraph of the introduction could be moved up before the objectives.

Methods

There should be a clear operational definition of Long COVID as a variable.

Results

The results show a low rate of Long COVID, and the bulk of the analysis focuses on severity, mortality, and quality of life — that is, on secondary objectives. The main objective stated — “This study aimed to investigate the factors associated with Long COVID among hospitalized patients during the predominance of the Delta variant” — is not adequately addressed (possibly due to sample size limitations).

The analysis and results regarding quality of life are well presented.

In summary, this is an interesting study that has improved compared to its previous version. It includes multivariate analyses; however, there is a lack of coherence between the stated main objective, the presentation of results, and the conclusions. It might be easier to restructure the order of the objectives than to rework the entire analysis to align it with the original main objective.

**Do you want your identity to be public for this peer review?** For information about this choice, including consent withdrawal, please see our Privacy Policy

Reviewer #1: No

Reviewer #3: **Yes: ** David Rene Rodriguez Lima

---

## [Author Response · Author response to Decision Letter 3]

5 Apr 2025

Dear Reviewers,

I have made efforts to revise the manuscript and address all the reviewers' comments, and I have submitted my responses in the 'Response to Reviewers' file through the system.

I sincerely appreciate all of your thoughtful comments and suggestions.

PIMPINAN KHAMMAWAN

---

## [Decision Letter · Decision Letter 3]

Severity, Mortality, Long COVID-19, and Quality of Life: Insights from a Cohort Study of Hospitalized COVID-19 Patients during the Delta Variant Predominance period in Thailand

PONE-D-24-33832R3

Dear Dr. khammawan,

We’re pleased to inform you that your manuscript has been judged scientifically suitable for publication and will be formally accepted for publication once it meets all outstanding technical requirements.

Kind regards,

Oyelola A. Adegboye, PhD

Academic Editor

PLOS ONE

Additional Editor Comments (optional):

Reviewers' comments:

Reviewer's Responses to Questions

**Comments to the Author**

Reviewer #1: All comments have been addressed

Reviewer #3: All comments have been addressed

2. Is the manuscript technically sound, and do the data support the conclusions?

Reviewer #1: Yes

Reviewer #3: Yes

3. Has the statistical analysis been performed appropriately and rigorously?

Reviewer #1: Yes

Reviewer #3: Yes

4. Have the authors made all data underlying the findings in their manuscript fully available?

Reviewer #1: Yes

Reviewer #3: Yes

5. Is the manuscript presented in an intelligible fashion and written in standard English?

Reviewer #1: Yes

Reviewer #3: Yes

Reviewer #1: (No Response)

Reviewer #3: Reviewer Report

Dear Editor,

Thank you for the opportunity to review this manuscript.

I appreciate the authors’ efforts in revising the paper. The manuscript has improved significantly. The objective is now clearly stated and better aligned with the methods and results presented. The suggestions and comments provided in the previous round of review were adequately addressed and corrected by the authors.

Given the improvements and the overall quality of the current version, I consider that the manuscript is now suitable for publication.

Sincerely,

Dr. David Rodríguez

Reviewer

**Do you want your identity to be public for this peer review?** For information about this choice, including consent withdrawal, please see our Privacy Policy

Reviewer #1: No

Reviewer #3: **Yes: ** David Rene Rodríguez Lima

---

## [Editor Report · Acceptance letter]

PONE-D-24-33832R3

PLOS ONE

Dear Dr. Khammawan,

I'm pleased to inform you that your manuscript has been deemed suitable for publication in PLOS ONE. Congratulations! Your manuscript is now being handed over to our production team.

Kind regards,

on behalf of

Assoc Prof Oyelola A. Adegboye

Academic Editor

PLOS ONE